# Effects of Different Exercise Conditions on Antioxidant Potential and Mental Assessment

**DOI:** 10.3390/sports9030036

**Published:** 2021-03-03

**Authors:** Kanaka Yatabe, Ryota Muroi, Takanori Kumai, Takashi Kotani, Shu Somemura, Naoko Yui, Yuka Murofushi, Fumiko Terawaki, Hajime Kobayashi, Kazuo Yudoh, Hiroyuki Sakurai, Hisao Miyano, Hiroto Fujiya

**Affiliations:** 1Department of Sports Medicine, St. Marianna University School of Medicine, 2-16-1 Sugao, Miyamae-ku, Kawasaki 216-8511, Japan; ryota.muroi@marianna-u.ac.jp (R.M.); aburainisuke@marianna-u.ac.jp (N.Y.); y.murofushi.cc@juntendo.ac.jp (Y.M.); f.terra@marianna-u.ac.jp (F.T.); fujiya-1487@marianna-u.ac.jp (H.F.); 2Department of Orthopaedic Surgery, St. Marianna University School of Medicine, 2-16-1 Sugao, Miyamae-ku, Kawasaki 216-8511, Japan; t2kumai@marianna-u.ac.jp (T.K.); t2kotani@marianna-u.ac.jp (T.K.); s3somemura@marianna-u.ac.jp (S.S.); 3Faculty of Health and Sports Science, Juntendo University, 1-1 Hiraka-gakuendai, Inzai, Chiba 270-1695, Japan; 4Health and Sports Science, Graduate School of Juntendo University, 1-1 Hiraka-gakuendai, Inzai, Chiba 270-1695, Japan; 5Orthopaedic Surgery, Yokohama Sports Medical Center, Nissan Stadium, 3302-5 Kozukue-cho, Kohoku-ku, Yokohama 222-0036, Japan; roadrunners56lb@yahoo.co.jp; 6Department of Frontier Medicine, Institute of Medical Science, St. Marianna University School of Medicine, Kawasaki 216-8511, Japan; yudo@marianna-u.ac.jp; 7Faculty of Psychology, Rissho University, 4-2-16 Osaki, Shinagawa-ku, Tokyo 141-8602, Japan; vyz03376@nifty.ne.jp; 8Department of Cognitive and Information Sciences, Faculty of Letters, Chiba University, 1-33 Yayoi-cho, Inage-ku, Chiba 263-8522, Japan; hisao.miyano@chiba-u.jp

**Keywords:** mental health, sports, sport psychology, exercise psychology, measurement

## Abstract

Exercise increases oxidative stress, leading the body to strengthen its antioxidant defenses, thus reducing the incidence of major diseases. As these associations are relatively unclear for ordinary levels of exercise for reduced stress, this study evaluated the effects of different exercise conditions on diacron-reactive oxygen metabolites (d-ROMs), biological antioxidant potential (BAP), and subjective mood. Forty-nine students (22.4 ± 2.6 years) were assessed using the Profile of Mood States (POMS) before and after exercising for 60 min. Participants were divided into two groups: Group A engaged in compulsory sports and Group B in freely chosen sports. d-ROMs and BAP were measured, and their modified ratio was calculated as an index of antioxidant potential. Physiological evaluation showed significant improvements in BAP and the BAP/d-ROMs ratio, irrespective of exercise condition (*p* < 0.001, *p* < 0.01). Comparison between the exercise conditions revealed a significant difference in the modified ratio (*p* < 0.02). In mood assessment, scores on emotion-related scales without vigor improved significantly under both exercise conditions (*p* < 0.001). Mental changes were evident after exercise, and potential antioxidant capacity was higher in freely chosen sports (*p* < 0.03). Assessment of antioxidant status before and after exercise may provide an objective index of mental and physical conditioning.

## 1. Introduction

Oxidative stress assessments are most often carried out in the event of illness. However, recent studies on the measurements of oxidative stress and biological antioxidant potential (BAP) in evaluating the condition of athletes have also started to emerge in the fields of sports and health sciences, in Japan and elsewhere [1,2,3,4,5,6]. Oxidative stress due to highly reactive oxygen species, such as hydrogen peroxide, damages deoxyribonucleic acid (DNA) and proteins, leading to deleterious effects in the body. Oxidative stress is induced in cells by means of a range of factors, such as external stress and oxidative phosphorylation in the mitochondria [7]. This may result in aging, the development of cancer, and a range of other diseases. However, reactive oxygen species are also believed to attack pathogens, thus functioning as part of the immune system. An appropriate level of stress only goes as far as inducing apoptosis, whereas excessive stress may lead to cell death [8]. Oxidative stress is evaluated through the assessment of hydroperoxide (ROOH) metabolites and other peroxides, quantified by returning them to radicals, while the BAP is a numerical expression of the reductive potency of FeCl_3_, which is used to assess sensitive reactions. The ΔBAP/d-ROMs ratio, obtained by dividing the diacron-reactive oxygen metabolites (d-ROMs) by BAP, is known as the antioxidant potential balance marker. There is a discussion on this process being adopted as a general index of oxidative stress in research and clinical practice in approximately 40 countries worldwide [9,10,11].

With respect to the assessment of oxidative stress, oxygen consumption is 10 times higher during exercise than at other times [12]. This causes a major increase in oxidized substances, which is one cause of post-exercise muscle fatigue. The muscle pain that occurs after vigorous exercise, particularly 24 h later, is also related to oxidative stress. Under normal circumstances, redox equilibrium in the body is maintained by the activity of non-enzymatic and enzymatic antioxidants. However, the amount of active oxygen and free radicals generated during vigorous exercise exceeds the body’s antioxidant defenses, inducing oxidative stress. The immune system’s response to exercise-induced damage is also believed to peak between 2 and 7 days after exercise. In terms of this relationship with exercise, studies have reported that during an endurance sport, such as 24 h mountain biking, exposure to long-term vigorous exercise induces an evident increase in antioxidant status (d-ROMs level) in athletes. This begins increasing 8 h after the beginning of the competition and continues throughout the race and for 24–48 h after the race [13]. Athletes exhibit low oxidative stress and high BAP, which may affect their performance. It has been found that during short-term intense exercise, the potential antioxidant capacity is increased and may provide a new evaluation index [14,15]. Following reports on the use of oxidative stress and antioxidant status in exercise challenge tests [16,17], a few case studies on competitive athletes have recently appeared in the literature. It has been found that the body strengthens its antioxidant defenses (particularly the glutathione system) to adjust to the exercise-induced increases in oxidative stress [18,19]. The risk of suffering from major diseases becomes lower in people who habitually exercise, an effect that appears to be helpful, to some extent, in preventing oxidative stress-related diseases [20].

At the level of exercise engaged in by ordinary people as stress relief, the associations between oxidative stress, antioxidant potency, and antioxidant potential balance markers, on the one hand, and psychological effects, on the other, remain unknown. Furthermore, an evaluation method is yet to be established. After very strenuous exercise, individuals’ heart rate, blood pressure, levels of stress hormones might be reduced although this physiological response may not appear for several hours. However, exercise can relieve mental stress whether or not it is related to this physiological response.

In this study, we explored the physical and psychological effects of exercise when ordinary people exercised for mental health purposes. We investigated potential changes in oxidation potential and psychological changes when ordinary individuals exercised for 1 h under several exercise conditions. We aimed to carry out simple measurements of the level of oxidative stress using the d-ROMs test and BAP. We also aimed to investigate the association between physiological evaluations and subjective mood assessments measuring oxidative stress before and after exercise to confirm whether exercise helps to relieve physiological and psychological stress (or improve mental health).

## 2. Materials and Methods

The participants were 49 medical students (32 men and 17 women; mean age, 22.4 ± 2.6 years). The exercise conditions comprised either a compulsory sport or a free choice of sport, with different conditions in each class in a sports medicine course, with Group A engaging in compulsory sports (fixed basketball, badminton, and table tennis, n = 25) and Group B engaging in freely chosen sports (number of sports not specified, n = 24) for 60 min. They engaged for 15 min each in three different sports and participated in a physical fitness test for 15 min (compulsory sport) or engaged in a sport of their choice (free choice). The effect of this exercise on their sense of mental well-being was then assessed. Peripheral blood was collected only from those who agreed and provided written informed consent to participate in the experiment. The participants’ peripheral blood was sampled during the elective class. We recruited participants from several years of elective classes. Before and after this 1 h of exercise, d-ROMs and BAP were measured (FRAS4, Wismerll Company Limited, Tokyo, Japan) and a mood assessment using the Profile of Mood States (POMS) was conducted [21,22]. Heart rate (HR) during exercise was measured with a Polar RS400 (Polar Japan, Tokyo, Japan), and the exercise intensities in Groups A and B were calculated.

### 2.1. Physiological Indices: d-ROMs and BAP Measurement

The physiological indices measured were d-ROMs and BAP. The current methods of evaluating the level of oxidative stress, which is quantification and numerical representation of oxidation and reduction in the body from measurements of peripheral blood samples, are the d-ROMs test (which measures oxidative stress in the form of reactive oxygen metabolites) and the BAP test. The d-ROMs test assesses ROOH metabolites and other peroxides, quantified by returning them to radicals, while BAP is a numerical expression of the reductive potency of FeCl_3_, which easily assesses sensitive reactions. The results of the d-ROMs test are expressed in Carratelli units (U. CARR), where 1 U. CARR corresponds to 0.08 mg/dL H_2_O_2_. Values higher than 300 U. CARR indicate oxidative stress. BAP levels are expressed as mol/L. The reference value provided by the manufacturer was >2200 mol/L. In this study, potential antioxidant capacity was calculated as the modified ratio obtained by dividing the BAP/d-ROMs ratio by a correction coefficient of 7.541 (modified ratio (potential: 1.0) = BAP/d-ROMs/correction coefficient 7.541) [23]. The level of improvement in potential antioxidant capacity was considered to have increased if the post-exercise value divided by the pre-exercise value was ≥1.0 and decreased if it was <1.0.

### 2.2. Mental Index: Subjective Mood Assessment

The POMS (Japanese version, Kaneko Shobo, Tokyo, Japan) was used for the subjective mood assessment [22]. This tool asks participants to rate their answers to the question “How do you feel right now?” for 65 items on a 5-point scale. It assesses six subscales: vigor, tension, depression, confusion, anger, and fatigue. Raw scores are transformed into standardized scores (T-scores), of which the median is 50 points. This profile is characterized by a high score on the vigor subscale and low scores on the other five subscales. The iceberg profile is characteristic of elite athletes and determined the profile of a successful athlete [24]. The pattern of mood responses in the iceberg profile was proposed to be reflective of positive mental health [25]. The overall mood was assessed in terms of a Total Mood Disturbance (TMD) score, calculated by subtracting the score for the positive subscale (vigor) from the combined score for the five negative subscales (tension, depression, confusion, anger and fatigue) [21].

### 2.3. Heart Rate Evaluation

An HR monitor (Polar RS400, Polar Japan, Tokyo, Japan) was used to measure the mean HR during exercise and to calculate predicted exercise intensity as a percentage of the theoretical maximum HR (HR max) [26].

### 2.4. Statistical Analysis

Statistical analysis was conducted using a single model with-in between interactions two-way analyses of variance (two-way ANOVA) and multivariate ANOVA. When there was a significant time (pre–post) interaction, differences in groups were evaluated where appropriate, with *p* < 0.05 regarded as significant. Data were expressed as means ± standard deviation (SD). The statistical software used was SPSS for Windows (24.0J, IBM Japan, Tokyo, Japan).

## 3. Results

In Group A, the mean HR was 132.3 ± 16.9 beats per minute (bpm), exercise intensity was 67.0%, and mean maximum HR was 186.0 ± 15.3 bpm. In Group B, the mean HR was 134.0 ± 12.7 bpm, exercise intensity was 65.1%, and mean maximum HR was 186.0 ± 14.3 bpm. Thus, participants from both groups engaged in almost equivalent levels of exercise intensity (60–70%) (Table 1).

This study has four oxidative stress variables (BAP; d-ROMs; ΔBAP/d-ROMs; modified ratio: BAP/d-ROMs/7.541 (Correction factor)) and two factors: time (pre–post), and group (A vs. B). The d-ROMs and BAP were not significantly different, but there was also interaction between Groups A and B from pre-test to post-test (Greenhouse-Geisser; d-ROMs, *p* = 0.129; BAP, *p* = 0.118). However, the ΔBAP/d-ROMs was significantly different between A group and B group from Pre to Post (Greenhouse-Geisser; *p* < 0.02) (Table 2).

Physiological evaluation of all the participants revealed significant improvement in the BAP and BAP/d-ROMs ratio, irrespective of the exercise condition (*p* < 0.001, *p* < 0.01). In Group A, BAP improved significantly post-exercise (*p* < 0.001). In Group B, both BAP and BAP/d-ROMs ratio improved significantly (*p* < 0.01). Although there was only a small change in the modified ratio, there was no significant difference between male (M) and female (F) participants whose values changed from before to after exercise in Group A. The change in the modified ratio was significantly greater in M participants than in F participants in Group B.

The modified BAP/d-ROMs ratio represents the antioxidant balance, which is calculated by dividing BAP by d-ROMs and then by 7.541. The higher this number, the more effective the protection from oxidative stress. The physiological assessment thus indicated that antioxidant potential improved significantly in Group B, irrespective of the different exercise conditions (A, pre-test, 0.89 ± 0.27; post-test, 0.94 ± 0.28; B, pre-test, 0.98 ± 0.35; post-test, 1.29 ± 0.64, Greenhouse-Geisser; *p* < 0.02; Table 2). The change in Group B became clear when it was calculated to the modified ratio. An investigation of the degree to which antioxidant potential improved (between participants increased (≥1.0) and decreased <1.0) in Groups A and B demonstrated the significance of d-ROMs, BAP, and antioxidant potential (Greenhouse-Geisser; d-ROMs, *p* < 0.01; BAP, *p* < 0.01; ΔBAP/d-ROMs, *p* < 0.001).

Additionally, this suggests that despite engaging in an exercise of similar intensity, those in Group B with a free choice of sports exhibited a greater degree of improvement in the modified ratio of antioxidant potential (Greenhouse-Geisser; *p* < 0.03). Their scores before and after exercise, although there were individual differences in the direction of change in Group A, an evaluation of their increases and decreases in potential antioxidant capacity demonstrated significant differences in d-ROMs, BAP, and the modified ratio. The Δ modified ratio of 17 participants (68.0%) improved in Group A, and 18 participants (75.0%) improved in Group B, giving a total of 35 participants (71.4%) who exhibited improvement. In Group B, there were individual differences in the direction of change, and the results were similar to those of Group A. However, a comparison of the two exercise conditions revealed a significant difference in the modified ratio (*p* < 0.03).

Most of our participants looked them a population with a normally low antioxidant potency at starting. The exercise intensity (predicted), despite being somewhat vigorous, could induce physiological changes, and there was no difference in the exercise conditions (Table 1). The corrected ratio also increased after exercise, although the overall increase was small.

### Mental Assessment

Mental assessment indicated that participants enjoyed an improved mood after exercise in both Groups A and B (Table 3). Significant differences were detected on all mood scales for the participants before and after exercise through repeated multivariate ANOVA (Wilks’ λ = 0.464, F (6, 40) = 7.708, *p* < 0.001). Vigor likewise increased, although the difference was not significant, and scores on the five negative subscales for emotions other than vigor (tension, depression, anger, fatigue, and confusion) all improved significantly (*p* < 0.001 in each case).

The difference between the two groups was evaluated using TMD. There tended to be interaction between Groups A and B from pre-test to post-test (Greenhouse-Geisser; *p =* 0.053). The TMD scores of both groups decreased significantly, showing that all participants’ moods were better after exercise (A, *p* < 0.001; B, *p* < 0.01; All, *p* < 0.001). It should provide a concise and precise description of the experimental results, their interpretation, as well as the experimental conclusions that can be drawn.

Finally, when we examined whether the changes in both antioxidant capacity and TMD before and after exercise were affected by the difference between the two groups, a significant difference was found (Wilks’ λ = 0.848, F (2, 46) = 4.118, *p* < 0.03). From the results of both physical and mental evaluation, mental changes were evident after exercise, and potential antioxidant capacity was higher in freely chosen sports.

## 4. Discussion

This study demonstrated the effect of exercise conditions on the antioxidant potential and mood states of healthy adults before and after engaging in compulsory sports (Group A) or freely chosen sports (Group B). Our study has shown the possibility examining to easily measuring the level of oxidative stress before and after exercise using the d-ROMs and BAP tests, to see if exercise helps improve mental health. At first glance, from the questionnaires Group A was seen as improvement in the mental plane. However, judging from both, the improvement in mental condition and the potential antioxidant capacity were greater after participating in Group B. Furthermore, assessment of antioxidant status before and after exercise may provide an objective index of mental and physical conditioning. Our findings highlight the importance of using objective indices of assessments for mental and physical conditions, such as examining an individual’s mood and antioxidant status before and after exercise.

Subjective mental assessment scores improved overall in both Groups A and B, with approximately 1 h of exercise, making their mood significantly more stable. This demonstrates that irrespective of the exercise condition, exercise performed at a certain intensity improves potential antioxidant capacity, has positive mental health effects, and may help to relieve stress. In addition, the level of improvement in antioxidant potential was significant in Group B for d-ROMs, BAP, and potential antioxidant capacity, showing a better psychological effect in the group where members were free to exercise at their own pace. This indicates that the same intensity of exercise strengthened the body’s ability to adjust to increased oxidative stress to a greater extent in those who were free to choose their own pace, type, and amount of exercise than in those who were asked to participate in a fixed exercise program [18,19]. This may help relieve stress and prevent diseases related to oxidative stress in people who habitually exercise for their mental health [20].

A limitation of this study was that the measurements had to be conducted within the class time and the oxidative stress levels in the peripheral blood, and the subsequent recovery could not be examined within this time frame. Of the class participants, only the small sample size was able to collect peripheral blood. For evaluation in the sports field, it is necessary to increase the number of participants and proceed with research from multiple angles. However, given individual differences in the effects on the immune, autonomic nervous, and endocrine systems, we think that d-ROMs and BAP measurements in the future might be appropriate for understanding in vivo redox abnormalities and assessing the in vivo balance within individuals in the sports field. Assessment of antioxidant status before and after exercise may provide an objective index of mental and physical conditioning. In the future, we should consider physiological index using saliva as simple physiological indices in addition to using questionnaires to evaluate psychological indicators.

## 5. Conclusions

More positive mental and physical changes were observed after 1 h of exercise wherein the pace was freely chosen by the individual. Our results suggest that an hour of exercise may increase potential antioxidant capacity and that the assessment of antioxidant status before and after exercise may provide an objective index of mental and physical conditioning.

Although the measurement of the d-ROMs/BAP ratio is yet to be widely used in the fields of sports and health sciences, it may potentially provide a simple physiological index for the evaluation of mental and physical conditioning in terms of physiological changes before and after exercise by individuals.

## Figures and Tables

**Table 1 sports-09-00036-t001:** Results of heart rate (HR) monitor (mean (± standard deviation)).

Group (n)		Resting HR (bpm)		AveHR (bpm)		HRMax (bpm)		ExStrength (%)	
A: fix (25) 22.2 ± 2.7 yrs.	M (13)	79.5 (19.5)	82.6 (17.2)	127.5 (14.1)	132.3 (16.9)	186.9 (16.5)	186.0 (15.3)	64.9 (7.9)	67.0 (8.9)
	F (12)	86.4 (14.0)		138.1 (18.8)		184.9 (14.5)		69.5 (9.7)	
B: free (24) 22.7 ± 2.5 yrs.	M (19)	83.2 (12.3)	82.3 (12.5)	133.7 (11.7)	134.0 (12.7)	187.6 (14.5)	186.0 (14.3)	67.7 (5.9)	65.1 (15.3)
	F (5)	78.8 (14.1)		135.8 (19.2)		178.3 (12.5)		69.1 (10.7)	

M: Men; F: Women; yrs.: years; bpm: beats per minute; AveHR: Average heart rate; HRMax: Maximum heart rate; ExStrength: Predictive exercise strength; A vs. B: *p* > 0.05 (All n.s.).

**Table 2 sports-09-00036-t002:** Results of diacron-reactive oxygen metabolites (d-ROMS), biological antioxidant potential (BAP) tests, and ΔBAP/d-ROMs ratio (mean (± standard deviation)).

Group		Item	Pre	Total	Post	Total
A: fix	M	d-ROMs	266.8 (65.5)	277.6 (67.5)	274.3 (71.0)	287.8 (77.5)
	F	(U. CARR)	289.3 (70.6)		302.3 (84.6)	
	M	BAP	1737.6 (240.7)	1753.9 (222.3)	1846.1 (131.8)	1894.5 (159.7)
	F	(μmol/L)	1771.5 (209.6)		1946.9 (175.8)	
	M	ΔBAP/d-ROMs	6.888 (2.010)	6.743 (2.044)	7.162 (1.977)	7.090 (2.109) *
	F		6.586 (2.158)		7.012 (2.330)	
	M	Modified ratio	0.913 (0.266)	0.894(0.271)	0.950 (0.262)	0.940(0.280) *
	F		0.873 (0.286)		0.930 (0.309)	
B: free	M	d-ROMs	229.9 (41.8)	244.6 (54.9)	219.3 (55.6)	237.1(64.7)
	F	(U. CARR)	300.4 (67.4)		304.8 (53.6)	
	M	BAP	1751.1 (534.1)	1732.8 (477.1)	2121.2 (582.3)	2055.1(535.7)
	F	(μmol/L)	1663.6 (131.6)		1804.0 (157.5)	
	M	ΔBAP/d-ROMs	7.811 (2.745)	7.383 (2.629)	10.712 (4.972)	9.751(4.830) *
	F		5.757 (1.293)		6.102 (1.376)	
	M	Modified ratio	1.036 (0.364)	0.979 (0.349)	1.420 (0.659)	1.293 (0.641) *
	F		0.763 (0.171)		0.809 (0.182)	

M: Men; F: Women; U. CARR: Carratelli units—an arbitrary unit of substance concentration expressed in milligrams per a volume of hydrogen peroxide. One Carratelli unit is equal to 0.08 mg H_2_O_2_/dL. Modified ratio: ={BAP/d-ROMs/7.541 (Correction factor)}; A vs. B (Pre-Post): * *p* < 0.02.

**Table 3 sports-09-00036-t003:** Profile of Mood States (POMS) (mean (±standard deviation)).

Scale	Group A: Fix		Group B: Free	
	Pre	Post	Pre	Post
Tension	47.66 (7.86)	42.68 (6.50) ***	43.81 (9.15)	41.16 (9.47) ***
Depression	50.41 (6.37)	44.38 (4.65) ***	49.05 (11.51)	45.56 (11.95) ***
Anger	47.42 (7.36)	40.45 (4.26) ***	46.55 (11.88)	41.88 (8.85) ***
Vigor	48.48 (9.55)	57.27 (9.95)	50.04 (9.47)	53.32 (12.41)
Fatigue	49.23 (8.02)	46.57 (7.75) ***	47.72 (10.46)	47.27 (11.18) ***
Confusion	49.98 (8.66)	44.58 (8.31) ***	46.95 (10.33)	42.68 (9.38) ***
TMD	134.88 (25.99)	109.96 (18.51) ***^,^ ^Δ^	126.13 (38.35)	112.63 (38.95) ***^,^ ^Δ^

TMD: Total Mood Disturbance, 6 scales (Pre-Post): *** *p* < 0.001, A vs. B (Pre-Post): ^Δ^
*p* = 0.053.

## Data Availability

The data associated with the paper is not publicly available due to them containing information that could compromise research participants’ privacy/consent, but may be obtained from the corresponding author on reasonable request.

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
