# Peer review of "Effects of Different Exercise Conditions on Antioxidant Potential and Mental Assessment"

_sports, 2021, doi:10.3390/sports9030036_

Round 1
Reviewer 1 Report
The study by Yatabe et al. has a simple design with interesting measures of oxidative stress after exercise. However, the article needs extensive revision. See specific comments below.
- The introduction is disconnected with the aims of the study. There are information the belongs to methods (L96-99), and no rationale to explain the relevance and novelty of acutely measure oxidative stress after exercise.
- The aims of the study (L100-103) do not mention that oxidative stress was measured before and after exercise.
- The aims (L103) state that the authors aim to “confirm whether exercise helps to relieve stress.” What kind of stress? Oxidative stress? Psychological Stress? Physiological Stress? Acute relief? Chronic relief? This reviewer believe that this aim goes far beyond the study design. For example, even after a very hard session of exercise (high intensity), the participant can experience lower levels of heart rate, blood pressure, and stress hormones. But this physiological response can take hours to show. This reviewer is not convinced that it is possible the measure “stress relief” immediately after the exercise session.
- Participants were assigned to group A or B by the research team, or through their sports medicine course? If it was the research team who made the assignment, it was made randomly?
- Figure 1 add nothing to the Results sections. The is not even a mention to Figure 1 in the results section.
- Please move your Tables and Figures following the paragraphs that refer to each one.
- Statistical analysis: the study has three oxidative stress variables (BAP; d-ROMs; BAP/d-ROMs) and two factors: time (pre-post), and group (A vs. B). When the authors use two models one t-test for within, and another t-test for between comparisons they are incurring in type 2 error. You should run a single model with within-between interactions (2-way ANOVA), considering your two factors mentioned above.
- Figures 2 and 4 do not show any data that are not already described in Table 4. If you’d like to keep the Figures, make a single figure with uniform plots for each oxidative stress marker. Or exclude all figures to keep the Table.
- Why HR is divided by sex?
- Please add a Table with characteristics and resting data by group, such as age, resting HR, number of females, etc. Then, use an independent t-test to compare the groups.
- The choice of graphic for Figure 3 is incorrect. Line graphics should be used when the independent variable (x-axis) is time related. However, time is being represented by different colors/lines and not by the x-axis. This graphic do allow this reviewer or reader to compare the values between groups (A vs. B).
- Everything in the Discussion section until it reaches L249, it is only rewording of what is written in the Results section. A proper and good Discussion section has to: state the main findings of the study; explain those findings based on previous literature; compare the results with similar literature; appoint the limitations of the study; show future directions; present some practical applications; then, conclude extrapolating the results and responding to the objective. However, the current Discussion do not cover any of these aspects.
Author Response
We thank referees for careful reading my manuscript and for your useful/helpful suggestions. Our responses to the first referee’s comments are as follow:
Response 1&2: We agreed with you and moved L96-99 to methods (NewL113-116). The aims of the study(L100-103), we have reflected this comment by mention (NewL91-105). We hope that you agree.
Response 3: We have incorporated your comments, we changed the expression of the last of introduction and tried to emphasize psychological and mental health (NewL94-105).
Response 4: We were assigned to group A or B through their sports medicine course by class. We added to explain this in detail (NewL109-118).
Response 5: We cut both Fig.1 and the explanation about four zone.
Response 6: We agree with you and have incorporated this suggestion throughout our paper. We did change in this time. (In this submission, position of the chart was specified here (3.3 Figures and Tables) by regulation, and it seemed that it would be included in the text when proofreading the block copy.)
Response 7: We agree with your suggestion. We changed the text in “2.4 Statistical analysis” and tried to change about this through all text. If you find any mistakes, please point out and give us guidance.
Response 8: We cut both fig. and change the table 2. We have four variables of oxidative stress and show not only the Δs but also the modified ratio in Table 2. We have rewritten overall the relevant part to be more in line with your comments. We hope that the edited section clarifies.
Response 9: Thank you for your comment. Rather than looking at the difference between men and women, it is generally said that the median heart rate of the resting heart rate differs between men and women. And since there are differences in exercise efforts this time, I dared to write them separately. If you still don't need it, we'll be ready to cut it.
Response 10: That is an interesting query. We added to data that we need more in Table 1.
Response 11: POMS (first edition) is intended to profile the 6 scale on a line graph for a long time, and is easier for readers to understand. We added to explain and references about this (NewL148-152), and changed from Fig 3 to Table 3 because there is too much data to convey.
Response 12: We tried to cut the explanation about four zones and move provide some clear explanations from the discussion to the results section, reflecting the opinions of all reviewers. We have rewritten in the discussion section to be more in line with your comments. We hope that the edited section clarifies. I hope it's communicated well, but I'm sorry I don't have the ability in English.
Again, thank you for giving us the opportunity to strengthen our manuscript with your valuable comments and queries. We have worked hard to incorporate your feedback and hope that these revisions persuade you to accept our submission.
Reviewer 2 Report
This is an interesting study regarding exercise and relief of stress by measuring the level of oxidative 100 stress in view of d-ROMs test and BAP. The authors found that mental conditions changed after exercise, and potential antioxidant capacity was higher in freely 41 chosen sports, where assessment of antioxidant status provided an objective 42 index of mental and physical conditioning before and after exercise.
Hence, this study aimed to carry out simple It also aimed to investigate the association between 101 physiological evaluations and subjective mood assessments before and after exercise to 102 confirm whether.
The study is interesting, and the adopted methods were reliable. I recommend accepting the manuscript.
Author Response
I thank referees for careful reading my manuscript and for your fruitful suggestions. Based on other’s peer-reviewed results, we have made some corrections to make the paper better.
Again, thank you for your kind many words our manuscript. We hope that these revisions persuade you to accept our submission.
Reviewer 3 Report
This manuscript describes a study that assessed BAP, d-ROM and emotional/mental state of male and female university students. Overall, the manuscript is very well written. The Introduction provides a comprehensive background and rationale for the study. The methods and results sections are clearly presented.
A separate section for figures is a little unusual, but results are presented clearly. I do not think you need Figure 1. It would be better to add axis labels and units to figure 4 and explain the quadrants and the whole figure clearly. You have also provided a clear explanation of Figure 1 at the beginning of the discussion section which is very helpful for the reader.
Section 2.3 Heart rate evaluation: Please add details to this sentence to improve clarity. "...exercise intensity as a percentage of theoretical HR max." Perhaps you can reference the Karvonen method, i.e. 220 - age = theoretical HR max.
The information in lines 96-99 should be included in the methods section. Although the authors mentioned before and after in line 102, they should make it clear that this applies to the measures mentioned in the previous sentence as well.
Clarify the type of stress in line 103.
Student group allocations are described clearly in my view.
Figure 1 is not helpful and should be removed as I mentioned previously. Please check all statistical analyses as it seems that t tests were used when ANOVAs would be appropriate.
I agree that having figures 2 and 4 is not helpful when the data is presented in the table and they should be removed.
I think comparing HR for sex is fine as long as it is justified in the methods and discussion.
Figure 3 should be corrected.
The Discussion does need a lot of improvement as it mainly restates the results and should be rewritten without this information.
The authors should make major changes to this manuscript before resubmitting.
Author Response
I thank referees for careful reading my manuscript and for your fruitful suggestions. Based on your peer-reviewed results, we have made some corrections to make the paper better.
Response 1: Thank you for this suggestion about Fig.1. We cut Fig.1 & Fig.4 and tried to change Table 2 as instructed by the reviewers. We tried to cut the explanation about four zone and move to provide some clear explanations from the discussion section to the results section. We hope that you agree.
Response 2: Thank you for next suggestion about Section 2.3 HR evaluation. We agree with you and add to explain theoretical HRmax and reference (NewL159-160).
Response 3&4: About Information in lines 96-99 & 103, we agreed with you and moved L96-99 to methods (NewL113-116). The aims of the study(L100-103), we have reflected this comment by mention (NewL91-105). We hope that you agree.
Response 5: About student’s group, thank you for agreeing. We were assigned to group A or B through their sports medicine course by class. We added to explain this in detail (NewL109-118) , reflecting the opinions of other reviewers.
Response 6: We cut fig.1, and changed the text in “2.4 Statistical analysis”. If you find any mistakes, please point out and give us guidance.
Response 7: We cut both fig2 & 4 and change the table 2. We have four variables of oxidative stress and show not only the Δs but also the modified ratio in Table 2. We have rewritten overall the relevant part to be more in line with your comments. We hope that the edited section clarifies.
Response 8: About HR for sex, thank you for agreeing (Rather than looking at the difference between men and women, it is generally said that the median heart rate of the resting heart rate differs between men and women. And since there are differences in exercise efforts this time, I dared to write them separately. If another reviewer still doesn’t need it, we'll be ready to cut it.)
Response 9: About Fig. 3, POMS (first edition) is intended to profile the 6 scale on a line graph for a long time, and is easier for readers to understand. We added to explain and references about this (NewL148-152). And we changed from Fig 3 to Table 3 because there is too much data to convey.
Response 10: About Discussion, we tried to cut the explanation about four zones and move provide some clear explanations from the discussion to the results section, reflecting the opinions of all reviewers. We have rewritten in the discussion section to be more in line with your comments. We hope that the edited section clarifies. I hope it's communicated well, but I'm sorry I don't have the ability in English.
Again, thank you for giving us the opportunity to strengthen our manuscript with your valuable comments and queries. We have worked hard to incorporate your feedback and hope that these revisions persuade you to accept our submission.
Round 2
Reviewer 1 Report
No further comments.